# SHAMANN: Shared Memory Augmented Neural Networks

## Abstract

Current state-of-the-art methods for semantic segmentation use deep neural networks to learn the segmentation mask from the input image signal as an image-to-image mapping. While these methods effectively exploit global image context, the learning and computational complexities are high. We propose shared memory augmented neural network actors as a dynamically scalable alternative. Based on a decomposition of the image into a sequence of local patches, we train such actors to sequentially segment each patch. To further increase the robustness and better capture shape priors, an external memory module is shared between different actors, providing an implicit mechanism for image information exchange. Finally, the patch-wise predictions are aggregated to a complete segmentation mask. We demonstrate the benefits of the new paradigm on a challenging lung segmentation problem based on chest X-Ray images, as well as on two synthetic tasks based on the MNIST dataset. On the X-Ray data, our method achieves state-of-the-art accuracy with a significantly reduced model size compared to reference methods. In addition, we reduce the number of failure cases by at least half.

## 1 Introduction

In the medical image analysis domain, the automatic parsing of medical images represents a fundamental task that impacts the efficiency of the entire clinical workflow from diagnosis to therapy planning, intervention and follow-up investigations. An essential step in this sense is the semantic segmentation of anatomical structures which supports the radiologist to read and understand the image content. Recent approaches are inspired from the vision domain and rely on fully convolutional neural networks, e.g., (Ronneberger et al., 2015; Yang et al., 2017), to achieve state-of-the-art results on various segmentation problems (Menze et al., 2015). Usually, these methods use the entire image to directly predict the complete segmentation mask. While this facilitates the incorporation of valuable global image context, it also increases the complexity of the learning task, requiring the models to capture the complete variability in the shape and structure of different objects. In addition, this strategy does not scale well to (volumetric) high resolution data due to memory limitations.

In this paper, we propose a new paradigm for semantic medical image segmentation based on a novel neural architecture called shared memory augmented neural network (SHAMANN). Based on a decomposition of the original image into a sequence of image subregions, e.g., local patches, we define different so called SHAMANN actors which traverse the sequence differently and segment each image subregion. An external memory module enables each actor to capture relations between different subregions within its sequence and increase the robustness of its predictions. In particular, this external module is shared among all actors and serves as a means to implicitly exchange local image context information in order to better capture global image properties, such as shape priors. Finally, the predictions of all actors are fused to obtain a semantic segmentation mask for the original image. An overview of the proposed framework with two SHAMANN actors is given in Figure 1.

The contributions of our work are: (i) a reformulation of the semantic segmentation problem as a sequence learning task (ii) SHAMANN - a memory efficient and dynamically scalabale alternative to end-to-end fully convolutional segmentation networks, that can also implicitly capture global image properties through a shared external memory module; and (iii) a comprehensive analysis of the method and comparison against state-of-the-art methods on a large chest X-Ray dataset.

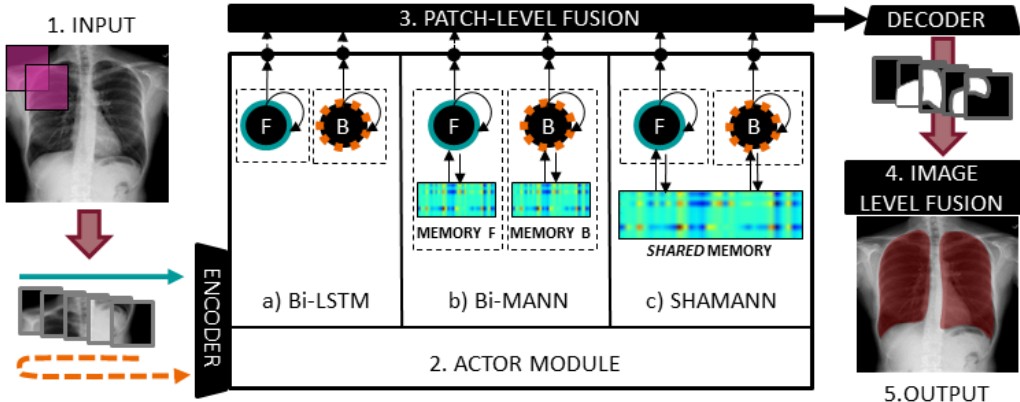

Figure 1: Generic architecture overview with two SHAMANN actors denoted by F and B that traverse a series of local image patches sequentially in a forward, respectively backward manner and segment each patch. For completion we also visualize two simplified alternatives, one in which the actors do not share their memory (Bi-MANN), and the other with no external memory (Bi-LSTM).

## 2 RELATED WORK

**Segmentation**. In the fields of computer vision and medical imaging, segmentation is a fundamental task for understanding the semantic content of an image. State-of-the-art results on different segmentation benchmarks (Cordts et al., 2015; Everingham et al., 2015), have been achieved by using fully convolutional neural networks (He et al., 2017; Shelhamer et al., 2017). However, one limitation of such networks is the use of pooling layers. By down-sampling and increasing the field-of-view, precise localization information is lost. To tackle this issue, two different approaches have been proposed. First, encoder-decoder architectures, e.g., U-NET (Ronneberger et al., 2015), recover the details and spatial dimension using de-convolutions and shortcut connections (Badrinarayanan et al., 2017; Lin et al., 2017; Yang et al., 2017). The alternative is to use dilated convolutions to increase the field-of-view without decreasing the spatial dimension (Chen et al., 2018; Peng et al., 2017; Yu et al., 2017; Yu & Koltun, 2015; Zhao et al., 2017). In this context, graphical models such as conditional random fields (Lafferty et al., 2001) are used to further improve the results. In the medical context, a standard approach for medical segmentation is multi-atlas label propagation (MALP) (Wang et al., 2013; Zikic et al., 2013). In MALP, a collection of atlases, i.e., labeled images, is required. At runtime, one needs to perform expensive non-linear registration operations of each atlas to unseen data to achieve a segmentation. These solutions typically scale poorly and are inefficient. Alternatively, one can address the segmentation problem by using random forests (Glocker et al., 2012), providing stronger unary predictions through joint class and shape regression. Milletari et al. (2017) employed an additional patch-voting scheme to increase the robustness against outliers. Other approaches use linear shape models to incorporate prior information (SSM) (Heimann & Meinzer, 2009). In marginal space deep learning (Ghesu et al., 2016), SSMs have been coupled with deep learning to enable the segmentation of anatomical structures. While these methods provide good results and are relatively easy to train, they do not exploit global anatomical information. In addition, the inference is time-consuming, especially for 3D images.

**Memory networks**. Recently, neural networks have been augmented with an external memory module to decouple the memorization capacity from the network parameters, making these methods better suitable for capturing long-range dependencies in sequential data. These networks have been used in the context of classification (Vinyals et al., 2016), meta-learning (Santoro et al., 2016; Sprechmann et al., 2018), reinforcement learning (Mnih et al., 2015; Pritzel et al., 2017), graph problems (Graves et al., 2016) or question answering (Graves et al., 2016; Sukhbaatar et al., 2015), to name a few. Closest to our work are generative methods (van den Oord et al., 2016), which model the conditional probability of a pixel based on previous pixels, using LSTMs. In contrast, we propose a sequence learning task for image segmentation and show that the memorization capacity can be improved using a shared external memory. Bahdanau et al. (2014) and Wang et al. (2016) proposed a memory-based strategy for the task of machine translation. They use a bidirectional RNN to encode

the input and save the concatenation of the outputs of the two units in a memory. After the sequence is processed and saved in the memory, a decoder reads from the memory and outputs the final predictions. In contrast, our proposed method allows information exchange between SHAMANN actors while processing the input sequence thereby enabling each agent to access global image context. To the best of our knowledge, this is the first paper that proposes a method based on memory networks for the task of image segmentation.

## 3 PROPOSED METHODS

In this section, we present our main contribution, the shared memory augmented neural networks (SHAMANN) architecture for semantic segmentation. Our observation is that in a bidirectional setup, information from different directions is not being explicitly exchanged. We hypothesize that by sharing an external memory, our networks can better capture global context, leading to a more accurate segmentation.

### 3.1 PROBLEM FORMULATION

In the following $x$ and $x^T$ will denote a row and column vector respectively, and $A$ a matrix. Following formulations are focused on but not limited to 2D images. Formally, let us consider an input image $I : \Omega \rightarrow \mathbb{R}^{H_I \times W_I \times C}$, with $\Omega \subset \mathbb{R}^2$ the image domain; $H_I, W_I$ and $C$ denoting the height, width and number of channels of the image signal. The goal of the segmentation task is to assign a label to every pixel/voxel $x$ in the input image, considering a predefined set of K object classes $\{y_1, \ldots, y_K\}$. The segmentation result can be represented as a set of segmentation channels $Y : \Omega \rightarrow \mathbb{R}^{H_I \times W_I \times K}$, where the value of a pixel $(x, y)$ of a given channel $k$ encodes the probability of observing the class $y_k$. A final segmentation mask can then be obtained by applying a softmax function along the different class-specific channels. In this work, we propose to reformulate the segmentation problem as a sequential learning task. Let us consider a sequence of T patches $P = \{P_0, \ldots, P_T\}$ covering the image domain, with $P_t : \Omega \rightarrow \mathbb{R}^{H_p \times W_p \times C}$, where $H_p, W_p$ are the height and width of the patch. For example, these patches may be extracted using uniform sampling. We propose to learn a function $f$ that maps the sequence of input patches to a sequence of patch segmentation masks as $f(P_t)_{t=0}^T = (\Phi_t)_{t=0}^T$, with $f : \mathbb{R}^{T \times H_p \times W_p \times C} \rightarrow \mathbb{R}^{T \times H_p \times W_p \times K}$.

### 3.2 ARCHITECTURE OVERVIEW

In this section, we introduce in more detail the components of our model (as can be seen in Figure 1). The encoder extracts a rich visual representation from the raw patch intensities. We model it as a function (e.g., a convolutional network), mapping the input to a $d$-dimensional feature space: $E(P_t)_{t=0}^T = (\psi_t)_{t=0}^T$, with $E : \mathbb{R}^{H_p \times W_p \times C} \rightarrow \mathbb{R}^d$. The actor module, defined as component 2, learns the sequence of input feature vectors $\Psi = \{\psi_0, \ldots, \psi_T\}$ and captures distal spatial dependencies. Each actor scans the input sequence $\Psi$ differently, to produce an output sequence $H^J = \{h_0^J, \ldots, h_T^J\}$, with $h^J \in \mathbb{R}^d$. Here, we use two actors, one scanning the input in the forward direction ($J := F$), and the other in the backward direction ($J := B$). The patch-level fusion step combines the outputs of the actors as $\sigma(H^F \oplus H^B) = H$, with $\sigma : \mathbb{R}^{2 \times d} \rightarrow \mathbb{R}^d$ and $\oplus$ the concatenation operator. The mapping $\sigma$ could be a simple function, e.g., an average or a concatenation operation. In our work, we propose to explicitly learn how to combine the different outputs using a neural network with a single fully connected layer. The decoder maps the fused outputs of the actors to patch segmentation masks as $D(h_t)_{t=0}^T = (\Phi_t)_{t=0}^T$, with $D : \mathbb{R}^d \rightarrow \mathbb{R}^{H_p \times W_p \times K}$. In the final image-level fusion step (see component 4), all patch segmentation masks $\Phi = \{\Phi_0, \ldots, \Phi_T\}$ are aggregated over the full image domain to generate the final segmentation mask $Y$. For fusion, we propose to use averaging (Iglesias & Sabuncu, 2015).

### 3.3 IMAGE SEGMENTATION AS A SEQUENTIAL LEARNING TASK

In the following sections, we show three different alternatives for the actor module. These are the bidirectional long-short term memory units (Bi-LSTM), described in Section 3.3.1; the bidirectional memory-augmented neural networks (Bi-MANN), described in Section 3.3.2; and our proposed SHAMANN framework (see Section 3.3.3).

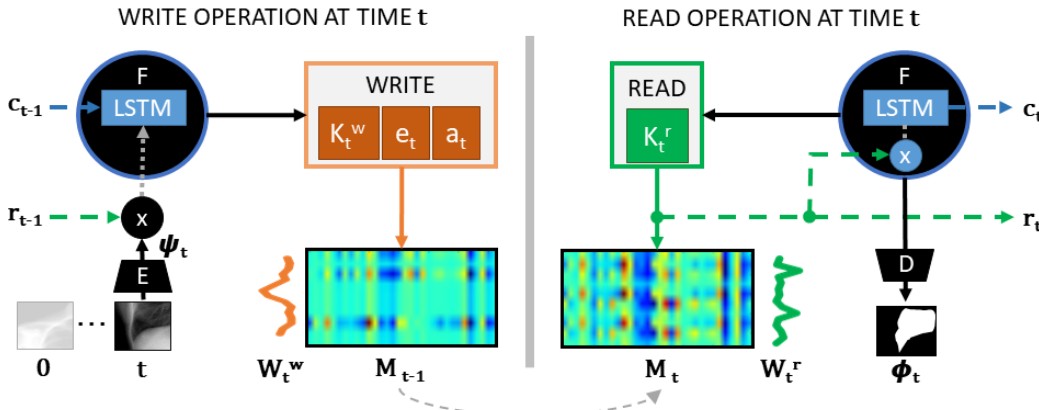

Figure 2: Detailed illustration of a forward actor network that uses an external memory module to perform a sequential segmentation task. At the time iteration $t$, the actor updates its memory cell $c_t$ using the previous memory cell $c_{t-1}$ and the current encoded input patch $\psi_t$, concatenated with the previous information read from the memory $r_{t-1}$. The actor then writes to and reads from an external memory module and produces a segmentation mask $\Phi_t$.

### 3.3.1 BIDIRECTIONAL LONG-SHORT TERM MEMORY NETWORKS: BI-LSTM

One of the most common challenge in training a recurrent neural netowrk is the vanishing gradient effect. To address this challenge, LSTM units have been proposed by Hochreiter & Schmidhuber (1997). These units have achieved high performance on real-world problems such as image captioning (Vinyals et al., 2015). The core element of the LSTM unit is the memory cell $c_t$, which is an abstract representation of the previously observed input. The definition of the output $h_t$ and $c_t$ can be summarized as:

$$[\boldsymbol{h}_t, \boldsymbol{c}_t] = LSTM(\boldsymbol{\psi}_t, \boldsymbol{h}_{t-1}, \boldsymbol{c}_{t-1}), \tag{1}$$

where LSTM stands for the gated processing structure. The output of a LSTM unit is the sequence of output vectors $H = \{\boldsymbol{h}_0, \ldots, \boldsymbol{h}_T\}$. The bidirectional LSTM processes the sequence data both in forward and backward directions with separate LSTM units. Thus, the forward LSTM unit will process the input sequence $\Psi^F = \{\boldsymbol{\psi}_0, \ldots, \boldsymbol{\psi}_T\}$ and produce the output sequence $H^F = \{\boldsymbol{h}_0^F, \ldots, \boldsymbol{h}_T^F\}$, while the backward LSTM cell will process the reverse input sequence $\Psi^B = \{\boldsymbol{\psi}_T, \ldots, \boldsymbol{\psi}_0\}$ and produce the output sequence $H^B = \{\boldsymbol{h}_T^B, \ldots, \boldsymbol{h}_0^B\}$. The final output of the Bi-LSTM is given by $H = \sigma(H^F \oplus H^B)$, where $\oplus$ denotes the concatenation operator.

### 3.3.2 BIDIRECTIONAL MEMORY-AUGMENTED NEURAL NETWORKS: BI-MANN

One limitation of Bi-LSTM is that the number of network parameters grows proportionally to the memorization capacity, making it unsuitable for sequences with long-range dependencies. These types of dependencies often occur in our formulation of the segmentation task, depending on the image content, the sequence length, and the patch size. One can alleviate this issue and increase the memorization capacity by making use of an external memory. These networks called memory augmented neural networks (MANN) use a controller network, i.e., an LSTM, to access an external, addressable memory $\boldsymbol{M} \in \mathbb{R}^{Q \times N}$, where $N$ is the number of memory cells and $Q$ is the dimension of each cell (Graves et al., 2016).

Following these principles, we propose to enhance each actor with an external memory capability. Figure 2 illustrates how a forward actor addresses such a memory module to perform a sequential segmentation task. At every time iteration $t$, the actor produces write and read heads to interact with a small portion of the memory constrained by weights associated with previous observations. The write operation uses the write weights $\boldsymbol{w}_t^w \in \mathbb{R}^N$ to remove content from the memory with an erase vector $\boldsymbol{e}_t \in [0, 1]^Q$, then write the add vector $\boldsymbol{a}_t \in \mathbb{R}^Q$: $\boldsymbol{M}_t[i] \leftarrow (\boldsymbol{1} - \boldsymbol{w}_t^w[i] \cdot \boldsymbol{e}_t) \circ \boldsymbol{M}_{t-1}[i] + \boldsymbol{w}_t^w[i] \cdot \boldsymbol{a}_t$, where $\circ$ and $\cdot$ denote the element-wise and scalar multiplication respectively and $\boldsymbol{1} \in \mathbb{R}^Q$ a vector of ones. Similarly, the output of a read operation using the read weights $\boldsymbol{w}_t^r \in \mathbb{R}^N$ is the weighted sum over the memory locations: $\boldsymbol{r}_t(\boldsymbol{M}) = \sum_{i=1}^N \boldsymbol{w}_t^r[i] \cdot \boldsymbol{M}_t[i]$. We

use content lookup to define the read weights, in which a key $\boldsymbol{k}_t^r \in \mathbb{R}^Q$ emitted by the actor is compared to the content of each memory location. The attention score for a read operation at row $i$ is the $i$-th value in the column vector $\boldsymbol{w}_t^r = exp(F(\boldsymbol{k}_t^r, \boldsymbol{M}_t[i])) / \sum_{j=1}^N exp(F(\boldsymbol{k}_t^r, \boldsymbol{M}_t[j]))$, where F computes the similarity between two vectors, i.e., cosine similarity, and [] is the row operator. The content lookup weights $\hat{\boldsymbol{w}}_t^w \in \mathbb{R}^N$ allow the write operation to update content in the memory. In order to also allocate new memory slots, we extend the addressing with a mechanism that returns the most unused location $\tilde{\boldsymbol{w}}_t^w \in \{0, 1\}^N$ (as a one hot vector). At every iteration the write operation uses an allocation gate $\alpha$ to either update the content of a location, or write to a new, unused location: $\boldsymbol{w}_t^w = \alpha \hat{\boldsymbol{w}}_t^w + (1 - \alpha)\tilde{\boldsymbol{w}}_t^w$. The read and write keys, erase and add vectors and the allocation gate are linear mappings of the memory cell of an actor.

We extend MANNs to a bidirectional formulation, where two actors, each with its individual external memory module, scans the input sequence in a forward ($J := F$) and backward ($J := B$) manner and produce the output and memory cell a time $t$ as:

$$[\boldsymbol{g}_t^J, \boldsymbol{c}_t^J] = LSTM(\boldsymbol{\psi}_t \oplus \boldsymbol{r}_{t-1}(\boldsymbol{M}^J), \boldsymbol{g}_{t-1}^J, \boldsymbol{c}_{t-1}^J), \tag{2}$$

where $\boldsymbol{g}_t^J = \boldsymbol{W}_g^J(\boldsymbol{h}_t^J \oplus \boldsymbol{r}_t(\boldsymbol{M}^J)) + \boldsymbol{b}_g^J$ are linear mappings of the concatenation of the output vectors and the currently read information from the memory module. The final output of Bi-MANN is given by $H = \sigma(\{\boldsymbol{g}_0^F, \boldsymbol{g}_1^F, \ldots, \boldsymbol{g}_T^F\} \oplus \{\boldsymbol{g}_T^B, \boldsymbol{g}_{T-1}^B, \ldots, \boldsymbol{g}_0^B\})$.

### 3.3.3 SHARED MEMORY-AUGMENTED NEURAL NETWORKS: SHAMANN

While the external memory module addresses the limited memorization capability of standard Bi-LSTM units, the sequence processing by the different actors remains suboptimal - in the sense that there is no active exchange of context information between them. The hypothesis is that through such an exchange, individual actors can observe more global context, leading to a more robust segmentation. With this in mind, we propose to share the external memory module between actors. By reading and writing information to the same memory module, the actors can interact in an implicit way. The output and memory cell for a time iteration $t$ are defined as follows:

$$[\boldsymbol{g}_t^J, \boldsymbol{c}_t^J] = LSTM(\boldsymbol{\psi}_t \oplus \boldsymbol{r}_{t-1}(\boldsymbol{M}), \boldsymbol{g}_{t-1}^J, \boldsymbol{c}_{t-1}^J), \tag{3}$$

where $\boldsymbol{g}_t^J = \boldsymbol{W}_g^J(\boldsymbol{h}_t^J \oplus \boldsymbol{r}_t(\boldsymbol{M})) + \boldsymbol{b}_g^J$ are linear mappings of the concatenation of the output vectors and the current read information from the shared memory module. Note that the matrix $\boldsymbol{M}$ in Equation 3 represents the memory module, which is shared by both the forward and backward actors, in contrast to Equation 2 where each actor has its own memory module, i.e., $\boldsymbol{M}^F$ and $\boldsymbol{M}^B$. The two actors write and read alternatively from the memory, first the forward actor, then the backward actor. To ensure the correct allocation of free memory, the two actors also share the usage vector. The final output of the SHAMANN framework is given by $H = \sigma(\{\boldsymbol{g}_0^F, \boldsymbol{g}_1^F, \ldots, \boldsymbol{g}_T^F\} \oplus \{\boldsymbol{g}_T^B, \boldsymbol{g}_{T-1}^B, \ldots, \boldsymbol{g}_0^B\})$. Our network is fully differentiable and can be trained end-to-end via back-propagation through time (Werbos, 1990).

## 4 EXPERIMENTS

In this section, we present the results of the proposed methods on real-world and synthetic applications. We benchmarked our method on a large chest X-Ray dataset and compared it to state-of-the-art methods. Additionally, we conducted two synthetic experiments on MNIST (Lecun et al., 1998) with the goal of analyzing the memorization capacity of the different models and providing insights into the benefits of sharing an external memory module.

### 4.1 CHEST X-RAY LUNG SEGMENTATION

This is a fundamental preprocessing task towards automated diagnosis of lung diseases, e.g., nodules, tumors, etc. (Wang et al., 2017). To meet high clinical standards, an accurate and robust segmentation of the lungs is required. For this problem, important challenges are the variability in shape and intensities of the lungs, as well as reduced anatomy contrast, due to pleural effusion.

The chest X-Ray dataset consists of 7083 images of 7083 patients selected from the public database ChestX-Ray8 (Wang et al., 2017), each of size $1024 \times 1024$ pixels. Ground truth segmentation masks

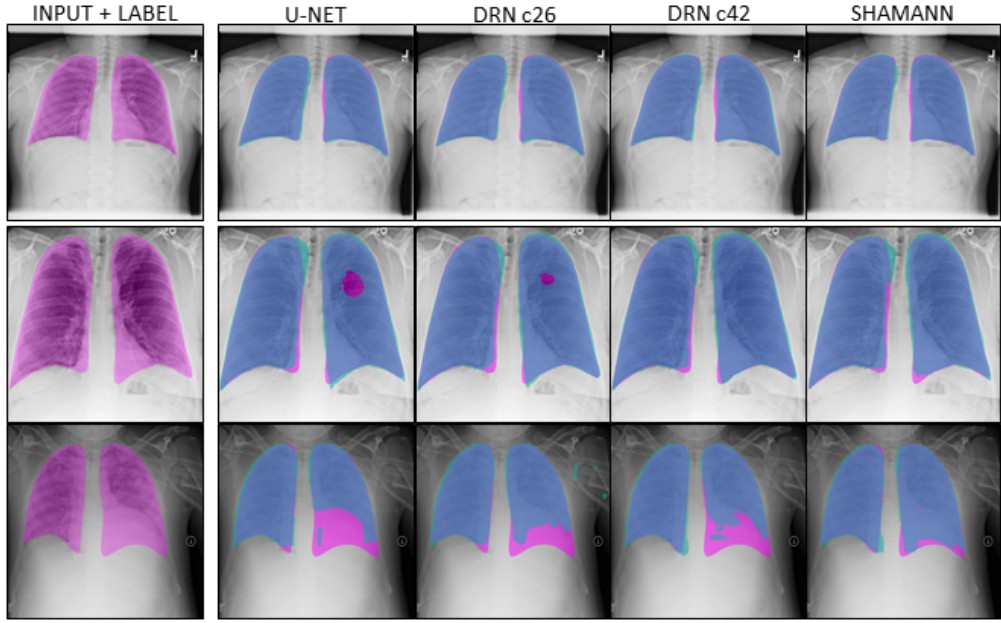

Figure 3: Qualitative results. From left to right: input image with label, segmentation masks using the method proposed by Ronneberger et al. (2015), two models proposed by Yu et al. (2017) and SHAMANN. The bottom two rows show results on more difficult cases. The last row demonstrates the effectiveness of the SHAMANN method in capturing the global context. We visualize the groundtruth mask in magenta, the prediction of the networks in turquoise and their overlap in blue.

Table 1: Quantitative results.

| Method | Dice | #params (millions) | high-res (3D) data | #failures |
|--------|------|--------------------|--------------------|-----------|
| **SHAMANN** | **96.97 +/- 1.36** | **6.2** | **flexible** | **5** |
| U-NET | 96.92 +/- 1.34 | 20.1 | memory limited | 10 |
| DRN(c26) | 96.95 +/- 1.67 | 20.6 | memory limited | 10 |
| DRN(c42) | 96.78 +/- 1.52 | 30.7 | memory limited | 12 |

were provided by clinical experts. We performed a random patient-based split in 5000 training, 583 validation and 1500 test images. The patch size was set to $160\times160$ pixels with a stride of $80\times80$, resulting in a sequence of 169 patches per image.

Table 1 shows quantitative results. We compute the dice score using the definition of true positive (TP), false positive (FP) and false negative (FN) as: $(2*TP)/(2*TP+FP+FN)$. The experiment demonstrates that, even though we use sequences of local patches, our algorithm reaches state-of-the-art performance by effectively capturing the global context through the shared memory. In particular, our model requires significantly less parameters in comparison to the reference methods. This allows in theory a more (memory) efficient application to high-resolution (volumetric) data. Furthermore, in our formulation one can dynamically split the sequence length (both at training and testing time) and maintain global context in the shared memory to achieve an even higher degree of flexibility. We are currently investigating these benefits on large volumetric medical scans.

An additional important property of our method is the robustness on difficult cases, caused by, e.g., large scale variations between children and adults, different image artifacts and abnormalities, such as pleural effusion or large lesions. We manage to reduce the number of cases with large error, i.e., a dice score below 0.9, by at least half. Figure 3 shows qualitative results.

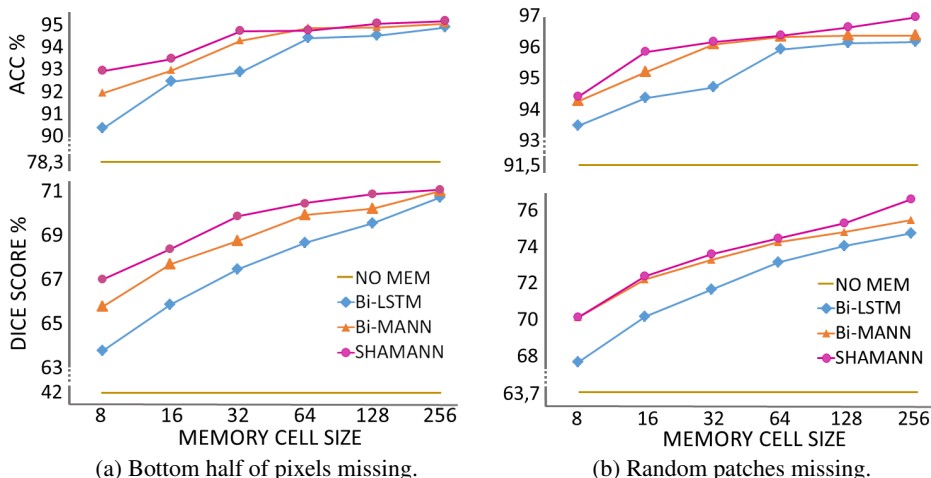

(a) Bottom half of pixels missing.    (b) Random patches missing.

Figure 4: MNIST quantitative results. SHAMANN performs best both in terms of dice and classification accuracy for different cell state sizes.

## 4.2 MNIST IMAGE COMPLETION: MEMORY ANALYSIS

To investigate the benefits of extending neural networks with an external memory, we designed two synthetic tasks based on the MNIST dataset. First, we deleted the bottom half of the input images and trained our models to complete the missing information. The goal of this experiment was to observe the networks capacity to extrapolate the missing data based only on the first half of the image. In a second experiment, we removed random patches from the input images. Since in this case the location of the missing data is not deterministic, the networks have to adaptively learn a more complex strategy for the memorization and lookup of information to better extrapolate the missing data. For both experiments we used the original MNIST images as labels. The MNIST data consists of 70000 pictures of handwritten digits (55000 train, 5000 validation and 10000 test) and their associated label. We considered patches of size 8×8 with a stride of 4×4 resulting in a sequence of 49 patches per image.

For the quantitative evaluation we measured dice scores, as well as classification accuracy on the reconstructed digits. To measure the classification accuracy we trained a deep neural network classifier on the original MNIST dataset and used this network to evaluate the images reconstructed by our methods. The accuracy of this classifier on the original MNIST dataset was **99.23%.** On the altered test sets, without applying any completion, the accuracy was **56.14%** for the first and **67.8%** for the second experiment. Figure 4 shows quantitative results. Using SHAMANN to perform image completion on the altered data, the classification accuracy was increased to **95.2%** for the first, and **96.9%** for the second experiment. In both experiments the networks augmented with memory outperform the Bi-LSTM network and especially the model without memory (called NO MEM). This demonstrates that more effective image completion strategies can be learned with the use of an external memory module, reaching best performance when the memory is shared. Note that as the capacity of the Bi-LSTM units increases, the difference in reconstruction performance to both Bi-MANN and SHAMANN reduces. As expected, given a large enough cell size, LSTM units can emulate the high memorization capacity of an external memory. While in the first experiment the methods perform similarly at the largest cell size; in the second experiment the differences between the methods is considerably large, even at the largest cell size level. This indicates that for more complex problems the performance of the Bi-LSTM is limited, even for a larger cell sizes.

Figures 5a and 5b show qualitative results. While in the first rows, the first three methods fail to extrapolate correctly the missing parts of the digits, the networks using a shared memory module make an accurate shape reconstruction that leads to correct classifications. The last row shows a failure case, where all four methods fail to correctly recognize the digit. However, considering the high difficulty in reconstructing these two digits, one can argue that the output of the SHAMANN method is reasonable. Figure 5c shows the benefits of sharing the memory module, by comparing

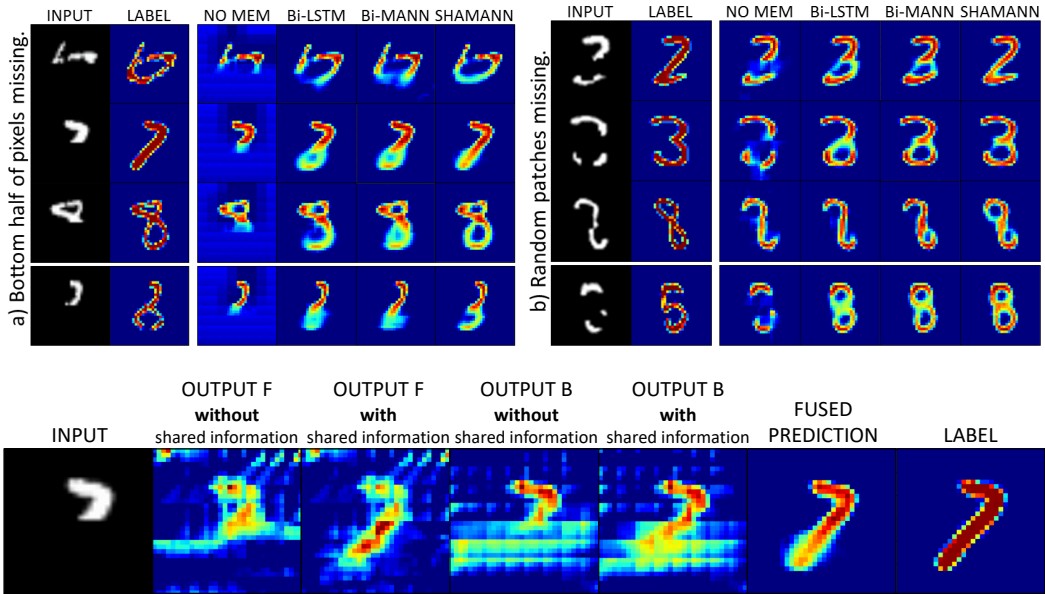

(c) Insights in the shared memory.

Figure 5: Figures 5a and 5b show qualitative results. From left to right: altered input image, label, reconstructed images using actors with no memory, actors with internal memory, actors with individual external memory, and actors that share an external memory module. Figure 5c illustrates how actors use the context seen by others to refine their prediction.

Table 2: Hyperparameters for both experiments.

|  | #layers | Encoder / Decoder shortcuts | #filters | Actor mem. cell | #read heads | Ext. Memory N | Q |
|---|---|---|---|---|---|---|---|
| X-Ray | 5 | yes | $\{8, 16, \ldots, 256\}$ | 128 | 32 | 400 | 128 |
| MNIST | 3 | no | $\{16, 16, 16\}$ | 8-256 | 8 | 100 | 8-256 |

the prediction of individual actors with and without the information exchange via the shared memory. Table 2 shows the hyperparameters used for the experiments. For training we used the RMSProp optimizer with a learning rate of $10^{-3}$ and minimized the mean squared error on all experiments.

## 5 CONCLUSION AND FUTURE WORK

In this paper, we presented a novel memory efficient segmentation approach based on sequence learning and memory augmented neural networks. Based on a decomposition of the original image into a sequence of image patches, we trained two SHAMANN actors that traverse the sequence bidirectionally and segment each image subregion. An external memory module enables each actor to capture relations between different subregions within its sequence and increase the robustness of its predictions. In particular, the shared nature of the external module serves as a means to implicitly exchange local image context information between actors to better capture shape priors. Despite the fact that we learn the segmentation module at patch-level, our algorithm matches the state-of-the-art performance of image-to-image architectures on a challenging lung segmentation task based on a X-Ray dataset. In addition, we conducted a detailed analysis on two synthetic tasks based on the MNIST dataset, demonstrating the benefits of sharing the external memory among different actors.

In our future work, we plan to extend our model to large 3D/4D medical scans and investigate the improved scalability and memory efficiency. We also plan to investigate the benefits of increasing the number of actors with different scanning strategies.

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
