# OpenReview forum: "SHAMANN: Shared Memory Augmented Neural Networks"
_ICLR.cc/2019/Conference_

### Official Review · AnonReviewer2 · 2018-11-02
**Interesting, but complex method for semantic segmentation. Unfortunately without convincing experimental results yet.**

**Rating:** 4
**Confidence:** 5

**Review:**

Summary:
The paper proposes a system of semantic segmentation based on sequential processing of the image in a patch-wise manner with multiple "actors", sharing a common external memory. This approach stands in contrast to the more usual approach of single-shot prediction for the whole image, where encoder-decoder architectures or dilated convolutions are used to capture the global context. The authors then discuss three-variants of this method, out of which two use external memory (Bi-MANN, SHAMANN), and one uses memory shared between actors (SHAMANN). Results are presented on segmentation of lung X-ray data and on MNIST digit completion.

Comments:
The paper is easy to read. The authors cite the relevant literature on the baseline semantic segmentation methods, as well as neural networks with external memories. However, similar patch-wise and sequential methods have been presented in the literature (e.g. https://arxiv.org/abs/1506.07452), including ones with external storage (e.g. https://www.nature.com/articles/s41592-018-0049-4), but these are not discussed as prior work.

Overall, the proposed approach is interesting, but significantly more complex than both the baselines and prior work. As is, the experimental results are not compelling enough to justify this (lack of clear quantitative improvement over state of the art). My recommendation would be to conduct additional experiments on semantic segmentation benchmark datasets. The proposed method seems promising for volumetric data as the authors note, but this also needs to be demonstrated experimentally.

Some more specific & technical questions follow:
- In Table 1, how is the confidence interval for the Dice score computed?
- Have any experiments been done with more than 2 actors?
- How exactly is the patch sequence formed, i.e. what is the spatial order of the patches? How much to the results depend on this order, if at all?
- In the discussion on page 6, it seems to be implied that the reduced parameter count should allow more efficient application to volumetric data. This is a bit surprising, since with modern networks it is usually the input size that is limiting, not the number of network parameters.
- Have experiments with Bi-MANN and Bi-LSTM been done on the X-ray segmentation data? How do the results compare to SHAMANN?
- How does the inference and training time compare to the baseline methods?

---

### Official Review · AnonReviewer1 · 2018-11-02
**paper seems well written and novel.**

**Rating:** 5
**Confidence:** 3

**Review:**

The authors present a model for semantic segmentation. The proposed method casts the full image segmentation as a sequence of local segmentation predictions. The image is split in multiple patches and processed sequentially in some order. A shared memory allows the local patch predictions to propagate information to improve other patch predictions which is necessary for resolving ambiguities.  They show a set of results on an XRay segmentation dataset with a reasonable ablation and baseline study. As well as a somewhat unclear result on image completion. The paper is well written, mostly clear and novel to the best of my knowledge.

pros:
- semantic segmentation is clearly very important problem with many applications
- the method seems clean and promising
cons:
- the segmentation community is much more familiar with MS-COCO and VOC. I think results on those datasets will make the paper much more impactful and clear any doubts about the method.
- it is not clear what processing order the patches are processed in. Does that matter ? This should be clearer in the paper.
- there is a brief mention of multiple actors but it seems to me its just one Bi-MANN actor is that true ?
- sec. 4.2 is very surprising to me. From what is written I understand that an MNIST classifier is trained on the original MNIST dataset and that it still works to 56% on the test set with the bottom blanked out. Is that correct ? What architecture is this ? Also I find it very surprising that you can recover accuracy to 96% without seeing the trained classifier at all. Anything that can help me understand how that is possible would be appreciated. Are you aware of anyone else matching these results in the literature ?

---

### Official Review · AnonReviewer3 · 2018-11-03
**lack of contributions and limited comparisons**

**Rating:** 4
**Confidence:** 5

**Review:**

The authors applied the external memory module proposed by Graves et al. (2016) to the image segmentation task. SHAMANN is an extension to allow memory sharing between directions.

Authors claimed that one of the contributions is a reformulation of the semantic segmentation problem as a sequence learning task.
There are many previous works done in this direction,
- "Multi-Dimensional Recurrent Neural Networks", 2007
- "Scene Labeling with LSTM Recurrent Neural Networks", 2015
- "ReSeg: A Recurrent Neural Network-Based Model for Semantic Segmentation", 2016
- "Robust, Simple Page Segmentation Using Hybrid Convolutional MDLSTM Networks", 2017
and many more.
Authors should compare with those LSTM-based image segmentation approaches as well.

Their second contribution is a network with a shared external memory module between directions. However, the experiments are not enough to show the benefits of it. See the details below.

Handling long-range dependencies:
- In Section 3.3.2, authors mentioned that "One limitation of Bi-LSTM is that the number of network parameters grows proportionally to the memorization capacity, making it unsuitable for sequences with long-range dependencies.".
However, the experiments are not with long range sequences: 169 sequence length for X-ray dataset and 49 length for MNIST. A classic LSTM (not bi-directional) is known to handle up to 200 timesteps. Some comparison/analysis of handling long-range dependencies of Bi-LSTM, Bi-MANN, and SHAMANN are needed (ideally on high-resolution real images).

Dataset:
-  Authors compared 3 models only on MNIST. The structure on MNIST is simple, and the resolution of images is small to show the benefit of using (shared) external memory module instead of individual memory cells. It is not surprising that the reported performance difference is small. Authors could have reported such a comparison on X-ray dataset too but they did not. I would recommend authors pick another high-resolution real-image dataset and compare the performance of these 3 models.

Additional comparisons:
- Various patch size
- Longer sequence length
- Especially a trade-off between the patch size and the sequence length on the high resolution images (larger patch size with a shorter sequence length or shorter patch size with a longer sequence length)

- A comparison of Bi-LSTM with sharing weights will also be a good baseline.

---

### Meta-Review · Area_Chair1 · 2018-12-14
**The paper can be improved**

**Confidence:** 4
**Recommendation:** Reject

**Metareview:**

The paper addresses the problem semantic segmentation using a sequential patch-based model. I agree with the reviewers that the contributions of the paper are not enough for a machine learning venue: (1) there has been prior work on using sequence models for segmentation and (2) the complexity of the proposed approach is not fully justified. The authors did not submit a rebuttal. I encourage the authors to take the feedback into account and improve the paper.